# MetaCOG: A Hierarchical Probabilistic Model for Learning Meta-Cognitive Visual Representations

**Marlene D. Berke**[1]    **Zhangir Azerbayev**[2]    **Mario Belledonne**[1]    **Zenna Tavares**[3,4]    **Julian Jara-Ettinger**[1,5]

[1]Psychology Dept., Yale University, New Haven, Connecticut, USA
[2]Computer Science Dept., Princeton University, Princeton, New Jersey, USA
[3]Basis Research Institute, New York, New York, USA
[4]Zuckerman Institute and Data Science Institute, Columbia University, New York, New York, USA
[5]Wu Tsai Institute, Yale University, New Haven, Connecticut, USA

## Abstract

Humans have the capacity to question what we see and to recognize when our vision is unreliable (e.g., when we realize that we are experiencing a visual illusion). Inspired by this capacity, we present MetaCOG: a hierarchical probabilistic model that can be attached to a neural object detector to monitor its outputs and determine their reliability. MetaCOG achieves this by learning a probabilistic model of the object detector's performance via Bayesian inference—i.e., a *meta-cognitive* representation of the network's propensity to hallucinate or miss different object categories. Given a set of video frames processed by an object detector, MetaCOG performs joint inference over the underlying 3D scene and the detector's performance, grounding inference on a basic assumption of object permanence. Paired with three neural object detectors, we show that MetaCOG accurately recovers each detector's performance parameters and improves the overall system's accuracy. We additionally show that MetaCOG is robust to varying levels of error in object detector outputs, showing proof-of-concept for a novel approach to the problem of detecting and correcting errors in vision systems when ground-truth is not available.

## 1 INTRODUCTION

Building accurate representations of the world is critical for prediction, inference, and planning in complex environments [Lake et al., 2017]. While the last decade has witnessed a revolution in the performance of end-to-end object recognition models, these models can nonetheless suffer from errors, particularly when confronted with out-of-sample data. How can we identify when a detected object is not actually there (a hallucination), or when an object present in a scene was not detected (a miss)?

Traditional approaches to this problem involve training the model to express intrinsic uncertainty over its outputs (e.g., Sensoy et al. [2018]; Kaplan et al. [2018]; Ivanovska et al. [2015]). While powerful, model performance can change when deployed in contexts not well represented in training. And sometimes, appropriate datasets for retraining are not always readily available. Here we focus on this class of problems. On these problems, the approach of post-hoc uncertainty learning (which seeks to extrinsically quantify uncertainty over a model's outputs) has the potential improve the reliability of a pre-trained off-the-shelf model (e.g., as in Shen et al. [2023]).

This paper draws inspiration from human cognition to introduce a proof-of-concept solution to the problem of post-hoc uncertainty learning. While human vision is generally robust and reliable, it nonetheless suffers from occasional errors, such as when we experience a visual illusion, like a mirage on a highway or apparent motion in a static image. Even though these illusions fool our visual system (i.e., they look real), people recognize that they shouldn't be trusted. This is because humans have internal models of how their own vision works that help them decide when to trust or mistrust what they see [Berke and Jara-Ettinger, 2024]—a form of *meta-cognition* [Nelson, 1990]. We propose that a similar approach can be used to improve the robustness of object detection models, by combining object detectors with an external model that represents the object detector's behavior (Fig. 1) and evaluates its outputs, integrating uncertainty and flagging unreliable detections. We present a formalization of this idea and a proof-of-concept of the approach.

There are two main sets of observations and insights that we draw upon. First, human vision is an encapsulated black-box system and we do not have access to its internal computations [Firestone and Scholl, 2016]. Yet, people are still able to learn a model of their own visual system [Berke and Jara-Ettinger, 2024]. This suggests that such learning is possible. We therefore focus on a problem formulation where

the meta-cognitive system does not have access to the internal architecture of the object detector—only its outputs. Second, people can detect failures in their visual system spontaneously and without need for feedback [Berke and Jara-Ettinger, 2024] and we therefore aim to do the same in our model. Our insight is that humans embed object detections in three-dimensional representations that assume object permanence (i.e., objects in the world continue to exist even when they cannot be seen; Carey [2009]; Spelke and Kinzler [2007]). For instance, if you detect an object in your periphery but it disappears when you look directly at it, you can conclude your visual system made an error, because objects do not spontaneously cease to exist.

Inspired by this, our model, MetaCOG (Meta-Cognitive Object-detecting Generative-model), aims to monitor an object detector's output and identify which detections were hallucinated and which objects were missed, without feedback or access to ground-truth object labels. MetaCOG achieves this by learning a model of the neural object detector's performance (i.e., a visual meta-cognition), represented as a joint distribution between objects in a scene and outputs produced by the object detector. Specifically, this probability distribution represents the object detector's propensity to hallucinate or miss objects of each visual category.

We evaluate MetaCOG's ability to 1) learn an accurate representation of the object detector's performance and 2) use this meta-representation to identify and correct missed or hallucinated objects. In Exp. 1, we explore MetaCOG's performance when paired with three modern neural object detectors, testing MetaCOG on a dataset of scenes rendered in the ThreeDWorld (TDW) virtual environment [Gan et al., 2020]. In Exp. 2, we explore MetaCOG's robustness by systematically testing its tolerance to different degrees of faulty inputs.

Our contributions are as follows: First, we propose a novel model, MetaCOG, that implements a human-inspired method for improving the performance of object detectors. Second, we introduce a new representation and inference method that enables accurate recovery of an object detector's propensity to miss and hallucinate object categories, without feedback or access to ground truth. Third, we show that, as a byproduct, MetaCOG's detection of misses and hallucinations allows us to automatically generate a tailored dataset of corrected errors, which may be used to fine-tune the object detector and improve its performance. Finally, we show through synthetic experiments that our approach is successful for a wide range of object detector performances.

## 2 RELATED WORK

**Meta-cognition in AI.** Previous work has shown the promise of meta-cognition for improving classification accuracy [Babu and Suresh, 2012, Subramanian et al., 2013].

While that work focused on engineering (rather than learning) a meta-cognition to guide training, we focus on a complementary problem: learning a meta-cognition for correcting outputs from a pre-trained network.

**Object knowledge.** Our work is also related to computational models of infant object knowledge [Smith et al., 2019, Kemp and Xu, 2009] and work applying cognitively-inspired object principles to computer vision [Chen et al., 2022, Tokmakov et al., 2021]. The difference is that MetaCOG uses object principles to learn a meta-cognition, whereas past work focused on modeling the object principles themselves.

**Neurosymbolic AI.** Our work can be seen as an instance of neurosymbolic AI [Garcez and Lamb, 2020]. Related work used detections from neural object detectors and a generative model of scenes to infer a symbolic scene graph [Gothoskar et al., 2021], but lacked the meta-cognitive component central to our work. In the language domain, Nye et al. [2021] used a symbolic world model to improve the coherence of a large language model, similar to how MetaCOG's symbolic world models and meta-cognition improve the coherence of object detections. Our work is unique in its formulation of meta-cognition within a neurosymbolic framework.

## 3 METACOG

Throughout, we consider the following problem formulation. Given a physical space (e.g., a room with objects in it), a set of images is sampled from a known camera trajectory $\vec{cj}$ (such that objects may pass in and out of view) and processed by an object detector. The goal of MetaCOG is to use this series of outputs from the object detector to learn about the detector's performance (i.e., what categories the detector is likely to miss or hallucinate), and to infer the underlying state of the physical space (i.e., the semantic label and 3D position of each object in the scene).

Fig. 1 illustrates MetaCOG's pipeline. Given a world state that is passed through an object detector, a list of object detections (which include their 2D location) is provided as input to MetaCOG. MetaCOG then uses Bayesian inference to jointly infer (1) a meta-cognitive representation and (2) the underlying world state (i.e., objects and their location in 3D space). The meta-cognitive representation is formed by two category-specific probability distributions: one capturing the probability of hallucinated detections (detected, but not actually there) for each category, and another capturing the probability of missed detections (not detected, but actually there) for each category. This representation is meta-cognitive in the sense that its content references not the external world, but the vision system itself. Due to its Bayesian nature, MetaCOG produces a posterior distri-

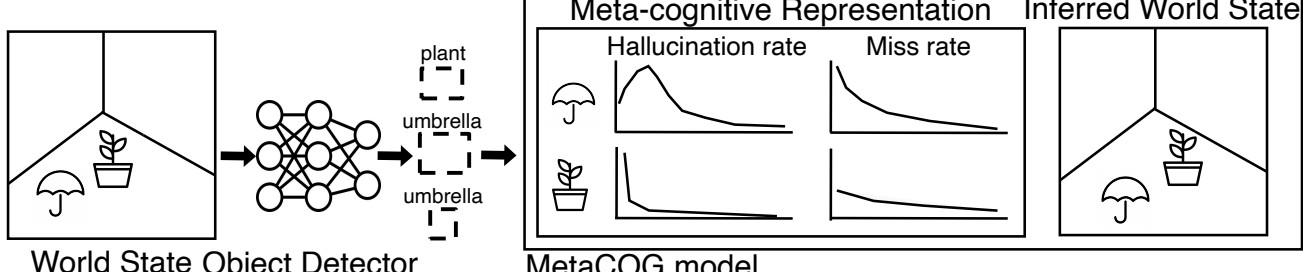

Figure 1: Conceptual schematic of MetaCOG. Information flows from left to right. From the left, images of a scene (described by a "World State") are processed by a neural "Object Detector," which produces detections (semantic labels with bounding boxes in 2D images). The MetaCOG model takes detections as input (without any access to the underlying world state or to the ground-truth accuracy of these detections), and jointly infers a meta-cognitive representation of the detector and what objects are where in the scene ("Inferred World State"). Specifically, MetaCOG's "Meta-cognitive Representation" consists of beliefs about the category-specific probabilities of the object detector generating hallucinations (detections of objects that were not actually there) and misses (failures to detect objects that were actually there). MetaCOG simultaneously infers this meta-cognitive representation and the world state (i.e., semantic labels and locations in 3D space).

bution over world states, and a posterior distribution over meta-cognitive representations. After repeating this process over multiple different world states, we can estimate the object detector's true probabilities of hallucinating and missing objects by taking the expectation of the posterior over meta-cognitive representations. The world states can be estimated using the MAP of the posterior over world states.

The rest of this section describes MetaCOG more formally, starting with how world states and detections are represented, building up to the meta-cognitive representation and the kernel for updating it, and finally concluding with the inference procedure.

### 3.1 GENERATIVE MODEL

**World states and object constraints** A world state is represented as a collection of objects, each object a 4-tuple of the form $(x, y, z, c)$ where $x$, $y$, and $z$ are coordinates in 3D space and $c$ is an object category. We denote by $\vec{W} = (W_1, \ldots, W_T)$ a sequence of world states.

Our approach implements object constraints into world states, which we hypothesize will make the visual meta-cognition learnable without access to ground-truth objects. Inspired by human infants' representations of objects, we assume that objects occupy locations in 3D space and that no two objects can occupy overlapping positions (implemented as a prior over the locations of the objects in a world state; details available in A.2). In this work, objects were assumed to be stationary, but the approach can extend to moving objects.[1] MetaCOG implements object permanence as the constraint that each world state has a fixed set of objects

that does not change (as a consequence, an object in a scene is still there even when it is not observed due to being out of view, occluded, or missed by the detector).

In our setting, multiple images are taken of each scene in a sequence of world states $\vec{W}$. We assume access to the camera's position and orientation for each image, which also implies knowledge of a scene change. While inferring the camera trajectory is possible [Taketomi et al., 2017], that is not the focus of our work. Critically, the MetaCOG model does not have access to the ground-truth objects in the underlying world states, and must instead infer them from detections output by a faulty object detector.

**Object detector outputs and inputs to MetaCOG** Given an image, a detector produces an unordered list of detections, each consisting of a category label and a position on the 2D image. Each detection is a tuple $(x, y, c)$ where $x$ and $y$ are pixel-coordinates (i.e., the centroid of a bounding box) and $c \in \mathbb{C}$ is an object category. The neural network object detector can be seen as a function $NN$ applied to an image $i$, producing a collection of detections, call this $NN(i)$. If $\mathbb{I}_{W_t}$ is the collection of images taken of the scene described by world state $W_t$, then the collection of detections generated from all of those images is $NN(\mathbb{I}_{W_t})$. For convenience, we will refer to this collection of detections generated from the world state $W_t$ as $D_t$. Crucially, while MetaCOG has access to the detections generated by a detector, the object detector itself is a black box—MetaCOG does not have any access to its internal state.

---

[1]The approach could be extended to moving objects by leveraging the human-inspired expectation that objects have spatio-temporal continuity, and therefore move in smooth and continuous

trajectories [Spelke and Kinzler, 2007]. This could be implemented as a prior over object motion. Rather than inferring a single location for each object, the problem becomes inferring a trajectory for each object.

**Meta-cognitive representation** The meta-cognition represents two aspects of the object detector's performance: its propensity to hallucinate objects that are not there, and its propensity to miss (or conversely, accurately detect) objects that are there. This is represented as two probability distributions per object category. The first distribution captures the detector's propensity to hallucinate, modeled as the number of times objects of category $c \in \mathbb{C}$ will be hallucinated in a given frame. Under naive assumptions, hallucinations may be independent of each other and randomly distributed within and across images, therefore following a Poisson distribution with rate $\lambda_c$ (to be inferred by MetaCOG). The second distribution captures the detector's propensity to correctly detect an object that is in view. Because object detectors can produce multiple detections from a single object (e.g., if it is incorrectly parsed as two objects), this representation follows a Geometric distribution with rate $p_c$ encoding a belief over the number of times $(0, 1, ..., k)$ an object of category $c$ will be detected when it is present in the image. Conveniently, under this formulation, the detector's miss rate for an object in category $c$ is $1 - p_c$.

Because each distribution is captured by a single parameter, the parameters of the distributions forming the meta-cognition can be represented as a pair of vectors of length $|\mathbb{C}|$ storing each category's hallucination rate $\lambda_c$ and miss rate $1 - p_c$. We call this pair of vectors of parameters $\theta$. We refer collectively to the distributions that they parameterize as $V$ since they express a belief about the veracity of the object detector. For notation simplicity, $V$ will refer to $V|\theta$.

The model described so far captures how MetaCOG represents an object detector's performance. However, a detector's propensity to hallucinate or miss objects can vary across scenes, so leaving flexibility in $V$ is desirable. At the same time, experience in a previous scene includes critical information about the detector that should inform expectations about its performance in a new scene. Our generative model therefore includes an evolving kernel, capturing changing beliefs over the parameters $\theta$ for the probability distributions $V$. Since beliefs about a detector evolve from scene to scene, $V_t$ denotes the belief at time $t$, and $\vec{V} = (V_1, ..., V_T)$ denotes a sequence of evolving beliefs over timesteps.

**Meta-cognitive learning kernel** The priors over the hallucination and miss rates are updated by matching up the objects that MetaCOG infers to be present in a scene with the detections that they caused. This allows MetaCOG to identify which objects were missed, and which detections were hallucinated. The details of the procedure for matching up objects and their detections are given in A.1.

Since the Gamma distribution is the conjugate prior of the Poisson distribution representing hallucinations, the Gamma distribution is used to model beliefs about each category's hallucination rate $\lambda_c$. At $t = 0$, the parameters of each

Gamma prior are initialized to $\alpha = 1$ and $\beta = 1$ (representing a relatively uninformative prior with a mean of 1). After inference over a scene, the Gamma prior over $\lambda_c$ evolves by computing the inferred number of hallucinations at time $t$ (based on the difference between world state $W_t$ and associated detections $D_t$, see A.1), and updating the $\alpha$ and $\beta$ parameters of the Gamma distribution.

Beliefs about the detection rates $p_c$ evolve in an analogous manner. As Beta is the conjugate prior of the Geometric distribution representing detections, beliefs about the miss rate evolve by updating the $\alpha$ and $\beta$ parameters of the Beta distribution. At $t = 0$, the Beta prior is initialized with parameters $\alpha = 1$ and $\beta = 1$ (equivalent to a uniform distribution on $[0, 1]$, a maximally uninformative prior for detection rate). The parameters from the Beta distribution are then updated based on the inferred missed detections (by comparing $W_t$ against $D_t$, see A.1).

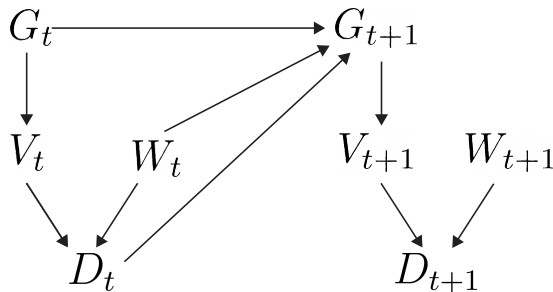

Figure 2: Schematic depicting the forward generative model of how world states $(W)$ and a vision system with some veracity $(V)$ produce detections $(D)$. $G_t$ is the prior over the meta-cognitive representation $(V_t)$ at time $t$; $W_t$ is the world state; and $D_t$ are the detections that are generated. $W_t$, $D_t$, and $G_t$ collectively influence $G_{t+1}$, the prior over $V_{t+1}$.

## 3.2 INFERENCE PROCEDURE

So far, we have described MetaCOG as a model of the generative process where 2D images of scenes are processed by an object detector to produce collections of detections. These detections, along with camera trajectories, serve as observations used to infer world states and a meta-cognitive representation of the object detector.

Given $\vec{D}$ (collections of detections $(D_1, ..., D_T)$ from a sequence of world states $\vec{W}$) and the corresponding camera trajectories $\vec{cj}$, the goal is to jointly infer (via Bayesian inference) the evolving meta-cognitive representation $\vec{V}$ and the world states $\vec{W}$ underlying each scene:

$$P\left(\vec{V}, \vec{W} \mid \vec{D}, \vec{cj}\right) \propto \prod_{t=1}^{T} P\left(D_t | W_t, V_t, cj_t\right) P(V_t) P(W_t)$$

The posterior is approximated via Sequential Monte-Carlo using a particle filter (for background, see Doucet et al. 2009). The details of the algorithms used for inference are described in A.3.

An estimate of the joint posterior can be sequentially approximated via:

$$
P\left(\vec{V}, \vec{W}|\vec{D}, \vec{cj}\right) \approx P\left(\hat{V_0}^0\right) *
\prod_{t=1}^{T} P\left(D_t|\hat{V_t}^t, \hat{W_t}^t, cj_t\right) P\left(\hat{W_t}^t\right) P\left(\hat{V_t}^t|\hat{W_{t-1}}^t, \hat{D_{t-1}}^t\right)
$$

where $\hat{W_1}^T, \ldots, \hat{W_T}^T$ is the estimate of $W_1, \ldots, W_T$ after detections from $T$ world states have been observed, and $\hat{V_1}^T, \ldots, \hat{V_T}^T$ is the estimate of $V_1, \ldots, V_T$ after $T$ observations. Here the transition kernel, $Pr(\hat{V_t}^t|\hat{W_{t-1}}^t, \hat{D_{t-1}}^t)$ is governed by the meta-cognitive learning kernel.

**Estimating $V$** After all $T$ world states have been processed, we estimate $V_T$ by taking the expectation of the marginal posterior distribution by averaging across particles weighted by their likelihood $l$: $\hat{V}_{T,\mu}^T = E[\hat{V}_T^T|\vec{D}] = \frac{1}{M}\sum_{m=1}^{M}(\hat{V}_{T,m}^T * l_m)$, where $m$ indexes the particles. This $\hat{V}_{T,\mu}^T$ is the final estimate of the belief about the true $V$ after all detections have been observed. Given $\hat{V}_{T,\mu}^T$, the conditional posterior can be estimated as:

$$
P\left(\hat{\vec{W}}|\vec{D}, \vec{cj}, \vec{V} = \hat{V}_{T,\mu}^T\right) \propto
\prod_{t=1}^{T} P\left(D_t|\hat{W}_t, cj_t, V_t = \hat{V}_{T,\mu}^T\right) P\left(\hat{W}_t\right)
$$

This conditional posterior can be used to infer world states for novel scenes, $W_{T+1}, ...$ or to reassess previous world states $W_1, ..., W_T$ that were originally inferred using a less informed meta-cognitive representation $\hat{V}$.

# 4 EXPERIMENTS

Our experiments have two goals: first, to test whether MetaCOG can learn an accurate meta-cognitive representation of an object detector, and second, to explore whether this meta-cognitive representation confers any benefits in overall accuracy or robustness to faulty inputs. Exp. 1 tests MetaCOG's performance when processing outputs of three popular object detection systems (Section 4.1). Exp. 2 presents a robustness analysis of how MetaCOG performs as a function of an object detector's baseline performance (Section 4.2). Throughout, we evaluate MetaCOG by sampling scenes (arrangements of objects in a room), and then sampling images taken from different viewpoints from each scene (calling the collection of images from a scene a "video"). Model code, datasets, results, and demos are available: https://github.com/marleneberke/MetaCOG/tree/main.

## 4.1 EXPERIMENT 1: ENHANCING NEURAL NETWORKS FOR OBJECT DETECTION WITH A META-COGNITION

**Object detection models** To test MetaCOG's capacity to learn and use a meta-cognitive representation, we tested its performance when paired with three modern neural networks for object detection. These networks represent three popular architectures: RetinaNet, a one-stage detector (Lin et al. 2017); Faster R-CNN, a two-stage detector (Ren et al. 2015); and DETR, a transformer (Carion et al. 2020), all pre-trained. The networks were validated to ensure that their baseline performances on our dataset were within their expected ranges. See B.2.2 for details.

**Dataset** We evaluate MetaCOG on a dataset rendered in the ThreeDWorld physical simulation platform (Gan et al. 2020) using one canonical object model per category. First, 100 scenes were generated by sampling objects and placing them in a room with carpeting and windows. For each scene, we then generated a video by sampling 20 frames from the ego-centric perspective of an agent moving around the room. The resulting set of 100 videos was then randomly split into a "training"[2] and test set (each with n=50 videos). To avoid order effects, all reported results show averages across four different video orders. See B.1 for the motivation for this dataset, sample images, and further details.

**Comparison models** We compare MetaCOG to two baselines in order to test if learning a meta-cognitive representation improves performance. First, we compare MetaCOG's inferences about world states to the neural network's post-processed detections serving as input to MetaCOG (see B.2.2 for details). Second, it is possible that the computational structure of MetaCOG might improve accuracy without meta-cognition per se mattering. For instance, mapping detections to 3D representations with object permanence alone might improve accuracy. Alternatively, having a meta-cognitive representation might provide tolerance for suppressing hallucinations or recovering missed objects, but the ability to infer it (and the resulting values of the parameters $\theta$ in $V$) might not matter. To test these possibilities, we compare our results against a *Lesioned MetaCOG*, where the values of $\theta$ were set to the mean of the initial prior over $\theta$ (see B.4.4). Although this model has a meta-cognitive representation $V$, the values of its parameters $\theta$ are neither learned nor updated in light of observations. Since this lesioned model contains the same representations as the full model, it serves as a control for model complexity.

---

[2]Note that the training set is not used for training in the traditional sense, since the ground-truth object labels are never used.

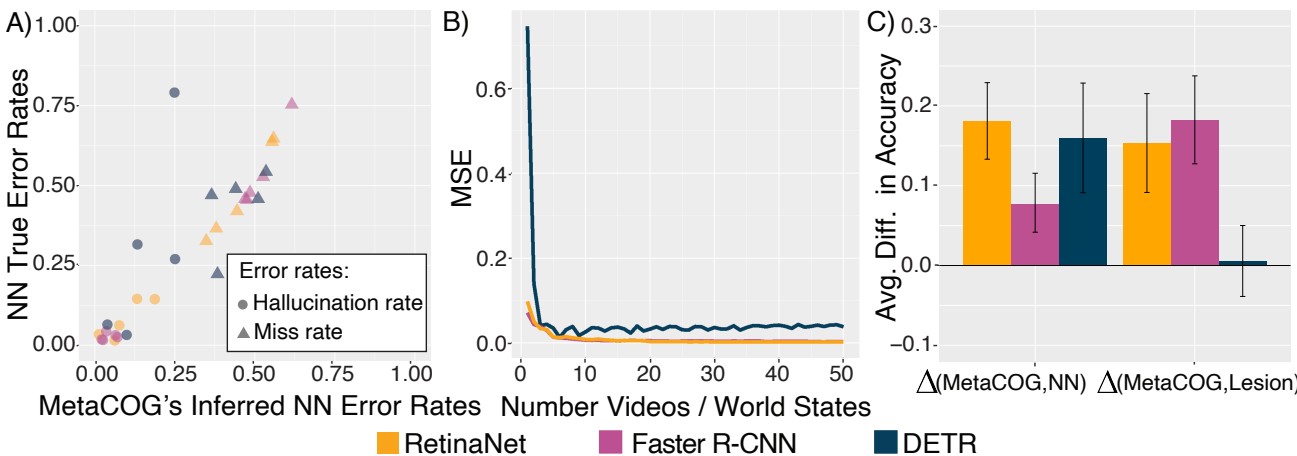

Figure 3: Results for MetaCOG and comparison models. Throughout, yellow codes for RetinaNet, magenta for Faster R-CNN, and blue-grey codes for DETR. A) Scatterplot showing MetaCOG's inferred values for the hallucination rates (circles) and miss rates (triangles) against the ground-truth values. B) The MSE (averaged across categories) of MetaCOG's inferences about $\theta$ as a function of number of videos observed. C) Comparisons between MetaCOG and the two baseline models on the test set (after conditioning on the meta-cognitive representation that MetaCOG inferred on the training set). The left group of bars show the difference between MetaCOG and the NN's output, and the right shows the difference between MetaCOG and the lesioned model. Positive values indicate MetaCOG outperforms the comparison model.

Table 1: Accuracy Results for Exp. 1. Values in Parentheses Are 95% Bootstrapped Confidence Intervals.

| OBJECT DETECTOR | MODEL | ACC. TRAINING | ACC. TEST |
|---|---|---|---|
| RETINANET | METACOG | **0.72** (0.66, 0.79) | **0.72** (0.65, 0.77) |
| | NN OUTPUT | 0.56 (0.52, 0.61) | 0.54 (0.50, 0.57) |
| | LESIONED METACOG | 0.64 (0.56, 0.71) | 0.56 (0.50, 0.63) |
| FASTER R-CNN | METACOG | **0.79** (0.73, 0.84) | **0.76** (0.71, 0.82) |
| | NN OUTPUT | 0.75 (0.70, 0.80) | 0.69 (0.64, 0.74) |
| | LESIONED METACOG | 0.66 (0.59, 0.73) | 0.58 (0.52, 0.64) |
| DETR | METACOG | 0.32 (0.25, 0.39) | **0.32** (0.24, 0.39) |
| | NN OUTPUT | 0.16 (0.13, 0.18) | 0.16 (0.13, 0.18) |
| | LESIONED METACOG | **0.36** (0.29, 0.44) | 0.31 (0.24, 0.38) |

**Results** We first assessed MetaCOG's ability to learn an accurate meta-cognitive representation $V$ (see B.3.1 for details). Fig. 3A shows the relationship between the parameters in MetaCOG's learned $\hat{\theta}$ and each network's true $\theta$, with an overall correlation of $r = 0.878$. Fig. 3B shows the learning trajectory, visualizing the MSE of $\hat{\theta}$ for each network as a function of experience (observed detections from videos). After 50 videos, MSEs decreased, on average, by 96.1% of their initial values, and had already decreased by 94.9% after only 20 videos. This demonstrates that MetaCOG can learn an accurate meta-cognition $V|\theta$ efficiently and without access to the ground-truth object labels.

Table 1 and Fig. 3C show MetaCOG's accuracy relative to comparison models (see C.3 for operationalization). MetaCOG outperformed the outputs of all three neural networks, and for two of the three, MetaCOG also outperformed the lesioned model. For DETR, we did not see a significant difference in accuracy between MetaCOG and

the lesioned version, perhaps because Lesioned MetaCOG was already performing well due to a coincidence where the priors over $\theta$ gave an adequate approximation of DETR's performance on this dataset (see B.4.5 for further discussion). When paired with Faster R-CNN, the full MetaCOG model improved the system's overall accuracy, but the lesioned model actually impaired performance. This is because the meta-cognitive representation in the lesioned model was a very poor description of Faster R-CNN's behavior. In particular, Faster R-CNN had very low hallucination rates ($\lambda_c < 0.05$ for every object category), while the lesioned model fixed all $\lambda_c$s to the mean of their priors ($\lambda_c = 1.0$). Because lesioned MetaCOG incorrectly represented Faster R-CNN as tending to hallucinate much more than it actually did, lesioned MetaCOG was overly skeptical of Faster R-CNN's detections and incorrectly dismissed many correct detections. This actually impeded overall performance, demonstrating the importance of MetaCOG's ability to learn and update its meta-cognitive representation

as a function of experience. Sometimes, a poor or misleading meta-cognition may be worse than no meta-cognition at all.

Overall, averaged across the three NNs, MetaCOG increased accuracy by 13.9% relative to the NN outputs in the test set. On average, MetaCOG also increased accuracy by 11.4% relative to Lesioned MetaCOG, confirming that MetaCOG's success can be partially attributed to its learned meta-cognition, rather than purely to the 3D object representation and priors (see B.4.1 for additional results showing that MetaCOG's inferences about the presence and locations of objects at the level of 3D scenes rather than 2D images also outperform those of Lesioned MetaCOG. Also see B.4.2 for results on an additional dataset).

While MetaCOG and Lesioned MetaCOG were matched in architecture, computational complexity, and input data, MetaCOG and NN Output were not. This raises the possibility that MetaCOG's success relative to the neural network's outputs could be due to its additional structure, computational complexity, and access to camera trajectories. To create a like-to-like comparison, we fine-tuned Faster R-CNN using MetaCOG's inferences on the training set (see B.4.3 for details) and compared it to off-the-shelf Faster R-CNN. It is possible to use MetaCOG to fine-tune an object detector since MetaCOG's inferences about what objects are where in 3D space can be projected back into 2D space to label the images, creating synthetic, labeled training data (all without access to ground-truth object labels). We compared Faster R-CNN's performance on the test set, before and after fine-tuning.

Although MetaCOG's inferences had an accuracy of only 0.76 (Fig. 4 MetaCOG), fine-tuning Faster R-CNN using MetaCOG's inferences improved the network's accuracy on the test set from 0.69 (Fig. 4 Off-the-shelf NN) to 0.81 (Fig. 4 Fine-tuned NN). Comparing Faster R-CNN before and after fine-tuning allows for a like-to-like comparison showing the impact of MetaCOG while controlling for computational complexity. Furthermore, these results serve as a proof-of-concept that MetaCOG can be used to train an object detection system.

Paired with this fine-tuned object detector, can MetaCOG use the NN's improved detections to make even better inferences? With more accurate inputs (0.81) MetaCOG did indeed draw even more accurate inferences (0.85; see MetaCOG Round II in Fig. 4). These results suggest that iteratively improving the object detector via fine-tuning and MetaCOG via better inputs has the potential to create a self-supervised learning system.

## 4.2 EXPERIMENT 2: ROBUSTNESS ANALYSIS

The results of Exp. 1 show that MetaCOG can efficiently learn a meta-cognitive representation for an object detector

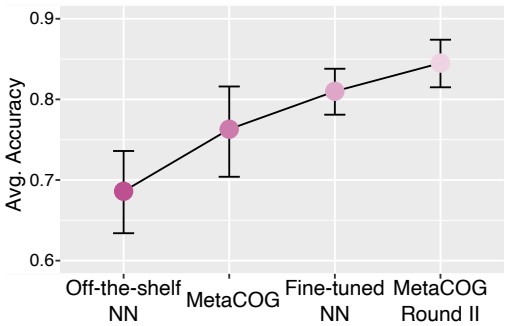

Figure 4: The results of MetaCOG fine-tuning Faster R-CNN. Each point shows the average accuracy of a model on the test set, and the bars are bootstrapped 95% CIs. From left to right, the leftmost point shows the accuracy of pre-trained, off-the-shelf Faster R-CNN. The next point shows the accuracy of MetaCOG, when paired with Faster R-CNN's inputs. The difference between these two points is depicted by the magenta Δ(MetaCOG, NN) bar in Fig. 3C. The third point shows the accuracy of Faster R-CNN after fine-tuning using MetaCOG's inferences. The rightmost point shows the accuracy of MetaCOG with inputs from fine-tuned Faster R-CNN. For exact accuracy values, see Table 5 in B.4.3.

and use it to improve detection accuracy. This provides proof-of-concept that MetaCOG can be applied to neural network object detectors. However, the three networks tested in Exp. 1 do not capture the full space of possible detector performances. How robust is MetaCOG to different degrees of error in its inputs?

To evaluate MetaCOG's robustness, Exp. 2 tests MetaCOG paired with a large range of possible simulated object detectors. To achieve this, we created a synthetic dataset of world states passed through simulated faulty object detectors, producing simulated detections. To reduce computational cost, we used an abstraction of world states and detections (removing all spatial components, detailed below) and used a lightweight, general-purpose version of MetaCOG (see C.1) to learn a meta-cognitive representation of the synthetic detectors and to infer world states.

**Dataset**  To focus on the contribution of meta-cognition for determining the presence or absence of different object categories in a way that is robust to the failures of the object detector, the dataset consisted of hypothetical collections of objects with no spatial information, passed through simulated object detectors with varying degrees of faulty performance. Specifically, we sequentially sampled world states (vectors of 1s and 0s indicating the presence or absence of five possible object categories), object detectors (miss and hallucination distributions $V$ with a wide variety of parameters $\theta$), and then generated faulty detections (vectors of 1s and 0s indicating the detection or lack of detection of each object category) by passing world states through the

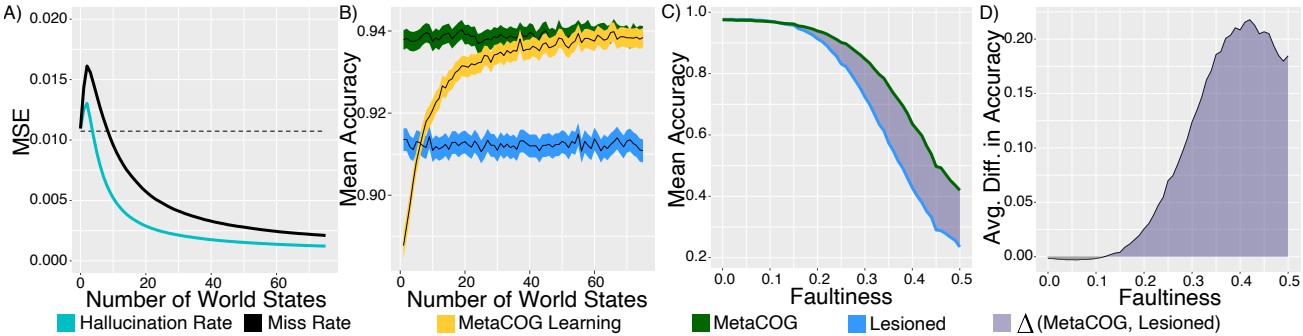

Figure 5: Lightweight MetaCOG's average performance over 40000 simulated detectors varying in faultiness, each processing 75 world states. **A)** MSE between true and inferred hallucination and miss rates as a function of number of processed world states. Horizontal dotted line represents the average MSE for the mean of the prior (the $\theta$ used in Lesioned MetaCOG). **B)** Mean accuracy as a function of number of processed world states. Average accuracy of (**C**) and difference between (**D**) MetaCOG and Lesioned MetaCOG as a function of faultiness ($\zeta$) in the detections.

probabilistic simulated object detectors. The final dataset consisted of 40000 randomly sampled object detectors, each processing 75 world states, with 5-15 processed frames (detection vectors) per world state. The details of synthesizing this dataset are left to C.2.

**Comparison models** As in Exp. 1, we compare MetaCOG to Lesioned MetaCOG, which fixes the parameters $\theta$ of the meta-cognitive representation $V$ to the expectation of the prior over $\theta$. This enables a fair comparison where the two models are matched for complexity. For details on the prior, see C.1, and for model details, see C.4.

**Results** Here, we demonstrate that, as in Exp. 1, MetaCOG can infer a detector's hallucination and miss rates without feedback, and that its improvements in accuracy track with its learning of a meta-cognitive representation. See C.3 for definitions of the metrics used.

Fig. 5 shows MetaCOG's performance over the 40000 sampled object detectors. To assess MetaCOG's ability to learn an accurate meta-cognitive representation $V|\theta$, we examined the MSE of $\hat{\theta}$ as a function of the number of observed videos. Fig. 5A shows that MetaCOG's estimates $\hat{\theta}_t$ of the hallucination and miss rates rapidly approach the true values $\theta$, with a final average MSE of 0.0017. To test if learning varied as a function of the object detector's errors, we next calculated MSE as a function of the detector's faultiness $\zeta$, defined as the detector's average proportion of errors per scene (see C.3 for formal definition). A linear regression predicting MSE as a function of faultiness revealed that MetaCOG's learned $\hat{\theta}$ is less accurate for faultier detectors ($\beta = 0.002$; $p < 0.001$), although the effect was minimal (predicting a MSE increase of 0.001 from a perfect detector with faultiness 0 to a detector with faultiness 0.5, where detections are equally likely to be true or false). Together, these analyses confirm that the results of Exp. 1 hold for a wider set of object detector performances.

Fig. 5B shows the average accuracy of the models' inferences about world states. The yellow line shows MetaCOG's rapid increase in accuracy over the first 40 world states. Over the course of these 40 world states, MetaCOG's accuracy increases from 88.8% to 93.6% (with 93.8% final accuracy after the 75th world states). This increase in accuracy occurs simultaneously with the decline in MSE for the inferred parameters of the meta-cognitive representation. By comparison, the blue line shows Lesioned MetaCOG's performance, which showed an average performance of 91.2%. The green line shows MetaCOG's accuracy after conditioning on the parameters $\hat{\theta}_T$ of meta-cognitive representation learned after observing the detections from all $T$ world states (see C.4 for details).

To quantify MetaCOG's robustness to faulty inputs, we next computed accuracy as a function of the object detector's faultiness $\zeta$ on a range from 0 (perfect performance) to 0.5 (detections equally likely to be true or false). Figure 5C shows MetaCOG's and Lesioned MetaCOG's mean accuracies as a function of $\zeta$. When faultiness is low, MetaCOG and Lesioned MetaCOG perform near ceiling, as the object detector's output is already highly reliable. However, MetaCOG outperforms Lesioned MetaCOG in accuracy for detectors with faultinesses in $\zeta \in [0.12, 0.5]$.

Fig. 5D shows the average difference in accuracy between MetaCOG and Lesioned MetaCOG as a function of faultiness. MetaCOG reaches its highest accuracy boost over Lesioned MetaCOG at faultiness level 0.42 (with a 21.8% improvement), and consistently shows a performance boost across a wide range of faultiness values. Together, these results show that MetaCOG's success is not limited to the particular neural networks tested in Exp. 1. Instead, MetaCOG can learn a good meta-cognitive representation for a wide range of detectors and use it to improve accuracy.

# 5 DISCUSSION

In order to behave intelligently in a complex and dynamic world, autonomous systems must be able to account for inevitable errors in perceptual processing. In humans, meta-cognition provides this critical ability. Inspired by human cognition, we formalize meta-cognition and apply it to the domain of object detection. Our work is a proof-of-concept of how a meta-cognitive representation can improve the accuracy of object detection systems. The MetaCOG model presented here can be directly applied to embodied systems, and MetaCOG may be particularly useful in situations where the vision system may be unreliable, like if the agent is deployed in an environment that was not adequately represented in training. Outside of this specific use-case, the more general MetaCOG approach can support robust AI more broadly.

First, the MetaCOG approach provides a way to quantify uncertainty, even for a pre-trained system and when ground-truth is not accessible. Sudden changes in the uncertainty expressed in the meta-cognitive representation could be used to detect domain or distribution shifts, so as to flag situations of high uncertainty that pose increased risk of unreliable behavior. This could be useful for downstream decision-making processes determining when to halt action.

Second, the MetaCOG approach can be used to tune and improve the underlying system that it is monitoring, as when MetaCOG's inferences were used for fine-tuning Faster R-CNN (Fig. 3). Using meta-cognition to generate a training signal is a novel alternative to human-annotation. In domains where the ground-truth labels required for traditional training are difficult to acquire, the MetaCOG approach could prove especially valuable.

Finally, our work focused on learning a meta-cognition for the purpose of improving accuracy and robustness, but the meta-cognitive representation also has potential benefits for transparency and interpretability. In our implementation, MetaCOG's $V$ captures the object detector's performance in a way that is easy for humans to understand. As such, this approach of learning a meta-cognitive representation may be able to support explainable AI by generating simplified meta-representations of a black-box system's performance.

# 6 CONCLUSION

We proposed a formalization of meta-cognition for object detectors that increases accuracy by removing hallucinations and filling in missed objects. Our model, MetaCOG, learns a probabilistic relation between detections and the objects causing them (thereby representing the detector's performance and instantiating a form of *meta-cognition*). This is achieved as joint inference over the objects and a meta-cognitive representation of the detector's tendency to error.

Critically, MetaCOG performs this inference without feedback or ground-truth object labels, instead using cognitively-inspired priors about objects. Applying MetaCOG to three neural object detectors (Exp. 1) and to simulated detectors (Exp. 2) showed that MetaCOG can efficiently learn an accurate meta-cognitive representation for a wide range of detectors and use it to account for errors, correctly inferring the objects in a scene in a way that is robust to the faultiness of the detector. This work is a proof-of-concept that meta-cognition can be used to quantify and correct a system's faults in the domain of computer vision and perhaps even beyond.

## Author Contributions

MDB developed the methodology, wrote the model code, conducted the experiments and analyses, and wrote the paper. ZA contributed to the code and methodology. MB provided technical and methodological advice (including the Random Finite Sets package in Julia) and edited the paper. ZT suggested experiments and helped with framing and writing the paper. JJE conceived the idea, supervised the project, and wrote the paper.

## Acknowledgements

This work was supported by NSF award IIS-2106690. We thank the Yale Center for Research Computing, specifically Tom Langford and Kaylea Nelson, for guidance and assistance in computation run on the Milgram cluster. Thanks to Aalap Shah, Eivinas Butkus, and Ilker Yildirm for helpful conversation and comments. We'd also like to acknowledge research assistants Ben Sterling, Bernardo Eilert Trevisan, and Tanushree Burman. Finally, thanks to Vatsal Patel for lending his sharp geometric reasoning!

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
