# OpenReview forum: "MetaCOG: A Heirarchical Probabilistic Model for Learning Meta-Cognitive Visual Representations"
_auai.org/UAI/2024/Conference — UAI 2024 poster_

### Official Review · Reviewer_D5Js · 2024-03-01

**Q2-1 Originality-Novelty:** 3
**Q2-2 Correctness-Technical Quality:** 3
**Q2-5 Clarity Of Writing:** 3

**Q1 Summary And Contributions:**

The paper proposes a model that helps to recognize the objects wrongly detected or missed by an object detection model. The proposed approach, MetaCOG, also helps fine-tune the underlying object recognition model. The main idea is to mimic the capability of humans to understand whether they are seeing real objects or having hallucinations. It is done by considering two distributions, one for detected object that actually are not present in the scene and one for objects present in the scene but not detected.

**Q2-3 Extent To Which Claims Are Supported By Evidence:**

3: Good: the main claims are supported by convincing evidence (in the form of adequate experimental evaluation, proofs, (pseudo-)code, references, assumptions).

**Q2-4 Reproducibility:**

4: Excellent: key resources (e.g. proofs, code, data) are available and key details (e.g. proof sketches, experimental setup) are comprehensively described for competent researchers to confidently and easily reproduce the main results.

**Q3 Main Strengths:**

The proposal is interesting and seems to achieve good results.
The approach seems to be easy to apply.

**Q4 Main Weakness:**

It is not clear how much it costs to apply the presented approach in terms of time and resources needed.

**Q5 Detailed Comments To The Authors:**

Overall, the paper is enjoyable to read. However, there are a couple of things that are not clear to me.

First, it is not clear if the objects in the scene can move, i.e., if the trajectories consider the movement of both the camera and the objects. This probably would not change either the formalization or the results, but I would suggest clarifying this aspect.

Secondly, I do not clearly understand how fine-tuning of the underlying object detection model is done. How the results of MetaCOG are incorporated into the loss function of the underlying model is not clearly discussed.

Then, it is not completely clear how much resources are needed to exploit MetaCOG. This could be important to define when and if the accuracy improvements are enough to justify the addition of MetaCOG in the pipeline of an application.

Lastly, I wonder why the lesionedMetaCOG's accuracy is worse than that of plain Faster R-CNN. The paper discusses the results of DETR but it tells nothing about the fact that the lesioned MetaCOG significantly decrease the performance of Faster R-CNN. This is probably a minor issue since the Lesioned version has less capacity, but I think the paper should at least highlight this aspect.

Typos:
- a parenthesis is missing at the end of the reference list of the 5th row of the second column on page 1
- in figure 2's caption, G_{t+1} is the prior over V_{t+1}

**Q9 Complying With Reviewing Instructions:**

Yes

---

> ### Author Rebuttal · Authors · 2024-04-05
>
> Thank you for your review! We are pleased that you found the approach to be “convincing” and recommended acceptance. We also appreciate your detailed comments, and will clarify those points below and in the manuscript. We hope that, upon clarification and discussion, that you might consider raising your score.
>
> *Clarification about object stationarity*
>
> In the setting considered, objects were stationary. As you note, our formalization and method should readily extend to moving objects. To extend to moving objects, the model would make use of the human-inspired prior expectation that objects have spatio-temporal continuity, and therefore move in smooth and continuous trajectories (Spelke & Kinzler, 2007). This would be implemented as a prior over object motion. So, rather than inferring a single location for each object, the problem becomes inferring a trajectory for each object. Thank you for pointing this out — we will clarify this in the revision and include this discussion of how to extend to moving objects.
>
> *Loss function for using MetaCOG to train Faster R-CNN*
>
> The details of fine-tuning the underlying neural networks in the Appendix, Section B.4.3 have now been expanded. Faster R-CNN is fine-tuned using its standard multi-task loss function given in the original Faster R-CNN paper (Ren et al., 2015, Eq. 1):
>
> L({p_i},{i_i}) = \frac{1}{N_{cls}} \sum{i} L_{cls}(p_i, p*_i)  + \lambda \frac{1}{N_{reg}} \sum{i} p*_i L_{reg}(t_i,t*_i)
>
> Where L_{cls} is the classification loss and L_{reg} is the regression loss. See Section B.4.3 of the updated appendix for further details. (Since we could not update the supplemental material directly on OpenReview, please see the revised Appendix uploaded to OSF via this anonymous link: https://osf.io/d4m8k/?view_only=9216bc5215a24e1f8c1e8faff218c655)
>
> MetaCOG’s inferences about what objects are where on each image are treated like ground-truth labels, so they are incorporated into the loss function as the ground-truth labels.
>
> *Computational costs/resources*
>
> We’ll start by noting that the goal of this work is to show an initial proof-of-concept that metacognition can be useful for object detection, and we were not optimizing for efficiency at this time. That said, we agree that it is important to consider the resources that MetaCOG uses.
>
> In Experiment 1, MetaCOG used 100 particle filters and 200 MCMC rejuvenation steps applied to each particle. We had access to 36 CPUs, so we were able to parallelize some of the particle filtering steps.
>
> Under this setup, when paired with Faster-RCNN, MetaCOG took an average of ~50 seconds per world state. There is a fair amount of variance (ranging from 20 to 102 seconds) per world state, owing to the complexity of the observations. To understand this variability, we have to delve deeper into the workings of the MetaCOG.
>
> The most computationally costly piece of MetaCOG’s pipeline is in the inference procedure: specifically, calculating the possible associations between detections and objects in the world. MetaCOG models both the object in the world states and the detections as random finite sets. The bottleneck is in calculating the likelihoods of different possible matchings between detections and objects in the world state. Under the semantics of random finite sets (Vo et al., 2013), the conditional distribution over a set of observations (in our context, detections) X given a hypothesis of elements E (in our case, putative objects) is calculated and marginalized over all partitions of E to X where each element in X is associated with an element in E. Since, in our case, the distribution of mass over partitions is sparse (i.e. most of the probability mass is concentrated in a small percentage of partitions), we can employ a random walk approach to substantially reduce runtime without impacting accuracy (Lovasz 1993; Blum et al., 2016) by reducing the complexity from O(N!) to O(N^2).
>
> Aside from the number of detections and putative objects in the world state, MetaCOG’s computation time depends on the amount of compute and the computational resources available. Our method uses a particle filter and MCMC rejuvenation steps applied to each particle. How many rejuvenation steps and particles are necessary for convergence depends on the dataset and the user’s tradeoff between speed and accuracy. In Experiment 1, we were optimizing for accuracy in the speed-vs-accuracy tradeoff, and so we erred on the side of overkill, using 100 particles and rejuvenating each particle using 200 MCMC steps. A use-case in which less compute is available could set a lower computational budget perhaps without compromising much accuracy. Extending the use-cases to situations in which speed is important requires studying how different computational budgets (i.e. different numbers of particles and MCMC steps) affect the speed-accuracy tradeoff.

---

### Official Review · Reviewer_WfRa · 2024-03-15

**Q2-1 Originality-Novelty:** 4
**Q2-2 Correctness-Technical Quality:** 3
**Q2-5 Clarity Of Writing:** 2

**Q1 Summary And Contributions:**

Inspired by human meta-cognitive capability to monitor his/her own cognitive system, the paper proposes a novel image recognition model equipped with a meta-cognitive module to evaluate and hence improve its own performance.  Empirical results show its effectiveness and robustness.

**Q2-3 Extent To Which Claims Are Supported By Evidence:**

3: Good: the main claims are supported by convincing evidence (in the form of adequate experimental evaluation, proofs, (pseudo-)code, references, assumptions).

**Q2-4 Reproducibility:**

3: Good: key resources (e.g. proofs, code, data) are available and key details (e.g. proofs, experimental setup) are sufficiently well-described for competent researchers to confidently reproduce the main results.

**Q3 Main Strengths:**

The approach using meta-cognition seems quite novel and interesting.  The experimental results about performance improvement and robustness seem reasonably convincing.

**Q4 Main Weakness:**

Although the proposed idea is fascinating, how it's achieved in the actual method is rather mysterious since details are not sufficiently and clearly described.  In particular, how the meta-cognitive module assesses the model's own performance and why it can do so are not so clear.  Also, even though their approach works in their particular setting, how it can be generalized to other settings is not clear.

**Q5 Detailed Comments To The Authors:**

Sect 3.1 gives their method but mostly by narratives.  As a result, how the whole model works is rather obscure, in particular, how meta-cognition works and why it can monitor the model's own performance by doing so.  I'm a bit worried about the particular choice of using the simple Poisson/Geometric distribution.  What is the principle behind the method?  Is it supported only empirically?  Overall, I think more maths should be used in the explanation and some more details given in the Appendix should be moved to the main text.  Also, the term "generative model" here is extremely confusing.

Discussion of related work is conspicuously missing.  The authors should do better to situate their work in a broader context, citing more related studies.

**Q9 Complying With Reviewing Instructions:**

Yes

---

> ### Author Rebuttal · Authors · 2024-04-05
>
> Thank you for your review! We are glad that you found the ​​idea to be “worth discussing” in UAI and that you recommended acceptance. We are pleased that you found the approach to be “quite novel and interesting,” the experimental results “reasonably convincing,” and rated the main ideas as “ground-breaking.”
>
> Thank you for your suggestions about how to improve the manuscript’s clarity. We will clarify our use of the term “generative model” — MetaCOG is generative in that it models the joint distribution over objects in the world (the targets of inference, Y) and detections (the observed outputs of the NN, X). Because MetaCOG models the joint P(X, Y), it is generative. Bayesian inference over this generative model is what enables MetaCOG to estimate Y by conditioning on X. We will clarify this in revision, or drop the term entirely.
>
> *How/why meta-cognition works*
>
> As you noted, some of the details and math of the model were left to the appendix (especially sections A.1, A.2, and A.3), which diminished the main body of the paper’s clarity on how the method works. We look forward to including these details in the main body of the paper for the camera-ready version, when space is no longer as constrained.
>
> Essentially, the meta-cognitive module (MetaCOG) is able to assess the performance of the underlying neural object detector by comparing the NNs detections against human-inspired principles of how objects work. Because MetaCOG assumes that objects persist in time and space and can’t overlap with one another (via spatio-temporal priors), MetaCOG is able to take the faulty detections $D_t$ from a scene and infer collections of objects $W_t$ that might produce that pattern of detections. By matching up the inferred objects to their detections (the details for this matching are given in A.2), MetaCOG recovers the number of detections not associated with an object (i.e., the “hallucinations”) and the number of objects not associated with a detection when they were in view (i.e., the “misses”). The number of hallucinations and misses in a scene are tallied separately for each object category.
>
>
> The tallies of hallucinations and misses for each object category are then used to update the priors over the hallucination and miss rates. Hallucinations are described as a Poisson distribution with rate parameter \lambda, and the conjugate prior of this distribution is a Gamma distribution with parameters (\alpha, \beta). Given a tally of hallucinations in a set of images, that incoming data updates the parameters of the conjugate prior — the use of conjugate priors provides an analytic way to update beliefs about the NN’s tendency to hallucinate. Beliefs about the miss rate are updated in an analogous manner using conjugate priors. This process for Bayesian belief updating (described in A.3) creates a dynamic “learning” process whereby beliefs about the NN’s hallucination and miss rate are refined from scene to scene, as a function of incoming information.
>
> We look forward to incorporating this into the main body of the paper once the camera-ready version affords us the space, and we think it addresses the mystery that you noted about how and why MetaCOG works.

---

### Official Review · Reviewer_dpkG · 2024-03-23

**Q2-1 Originality-Novelty:** 2
**Q2-2 Correctness-Technical Quality:** 3
**Q2-5 Clarity Of Writing:** 4

**Q1 Summary And Contributions:**

The authors propose a hierarchical probabilistic model designed to enhance the reliability of neural object detectors. They consider a setting where a scene is observed through a sampled set of images from a known camera trajectory. These sampled frames are given as input to object detectors, and the corresponding outputs are used to build a representation. This representation consists of two probability distributions over the categories, where one describes hallucinations, and the other focuses on missed detections. Then, they perform joint inference over this representation and the state of the scene.

**Q2-3 Extent To Which Claims Are Supported By Evidence:**

2: Fair: the main claims are somewhat supported by evidence (but the experimental evaluation may be weak, or does not match entirely with the claims, important baselines may be missing, proofs contain important ideas but lack rigor, algorithmic details are only discussed superficially, references are imprecise, assumptions are not sufficiently motivated or explicated, etc.).

**Q2-4 Reproducibility:**

3: Good: key resources (e.g. proofs, code, data) are available and key details (e.g. proofs, experimental setup) are sufficiently well-described for competent researchers to confidently reproduce the main results.

**Q3 Main Strengths:**

Overall, the idea seems interesting; it also provides a way to quantify the uncertainty of the underlying neural systems.

**Q4 Main Weakness:**

1. No experiments on real-world datasets.
2. No experiments on standard 3D datasets.
3. It is hard to validate the efficacy of the method.

**Q5 Detailed Comments To The Authors:**

Although the idea proposed is interesting, my main concern lies in the experimental validation of this method.

**Q9 Complying With Reviewing Instructions:**

Yes

---

> ### Author Rebuttal · Authors · 2024-04-05
>
> Thank you for the time and consideration that you spent reviewing our submission. We are glad that you found the idea to be an interesting way to quantify the uncertainty of neural systems.
>
> The main weakness that you identified was that the experiments were not conducted on real-world or standard 3D datasets. In general, real-world and standard 3D datasets are very useful for establishing model efficacy. However, in this work, we present a new and distinct kind of model and an initial test of whether, in principle, a meta-cognition for object detection could be learned and used to improve detections. In this initial stage, proof-of-concept testing required evaluation on a dataset with specific properties that we could control.
>
> Our goal was not to show MetaCOG works on real-world datasets, but to establish the efficacy in tightly controlled scenarios. Rendering custom datasets enabled us to very precisely understand the robustness of the system. For example, in the additional experiment on another dataset rendered in ThreeDWorld (see Appendix section B.4.2), we were able vary the crowdedness and complexity of the scenes (and subsequently, occlusion) by adding more objects into smaller and more varied rooms. To systematically vary backgrounds and crowding while controlling all other variables would have been impossible using real-world or standard 3D datasets. Furthermore, with real-world or standard datasets, we would be unable to control the experimental set up in the way that allowed us to prove the method’s robustness in Experiment 2. We believe that the present experiments were sufficient to show proof-of-concept.
>
> This follows a long tradition of developing statistical methods in controlled simulated environments to understand the properties of the framework, which can then be transferred into real-world cases. This is why we think UAI is an ideal venue, since it has long supported this type of work. For instance, in last year alone, UAI published important contributions by Bhattacharjee et al., Kang & Kim et al., Oesterheld et al., Piwek et al., and Lippe et al., among many others, all using only custom, synthetic datasets and/or simulations experiments. The latter (Lippe et al., 2023) even earned a spotlight, and its experiments used simulated environments very similar to ThreeDWorld. It is also worth noting that some of the most significant contributions to AI, such as GANs, began by establishing the system’s robustness in controlled, simulated settings.
>
> Given that our goal is to rigorously test whether this method can work under tight parametric variations, it would be helpful to understand the specific concerns about using simulated datasets. We don’t believe that we have made any claims that specifically require testing on a real-world dataset, and if we have, we will curtail our claims to those that are supported by the present experiments. As it stands, we believe that the present experiments are sufficient to support a proof-of-concept of the promise of meta-cognition.

---

### Meta-Review · Area_Chair_TZ75 · 2024-04-16

Despite some doubts about the empirical performance on real data, reviewers acknowledge the important and innovative idea of generating insights about the confidence of object detection.